# Infant Death Clustering in the Quarter of a Century in India: A Decomposition Analysis

**DOI:** 10.3390/ijerph192114384

**Published:** 2022-11-03

**Authors:** Mukesh Ranjan, Laxmi Kant Dwivedi, Shivalingappa Halli

**Affiliations:** 1Department of Statistics, Mizoram University, Pachhunga University College Campus, Aizawl 796001, Mizoram, India; 2Department of Survey Research & Data Analytics, International Institute for Population Sciences, Govandi Station Road, Deonar 400088, Mumbai, India; 3Department of Community Health Sciences, Institute for Global Public Health, Rady Faculty of Health Sciences, University of Manitoba, Winnipeg, MB R3E 0T6, Canada

**Keywords:** infant death clustering, national family health survey, random effects dynamic probit model, decomposition analysis, infant mortality rate

## Abstract

The study aims to examine the clustering of infant deaths in India and the relative contribution of infant death clustering after accounting for the socio-economic and biodemographic factors that explain the decline in infant deaths. The study utilized 10 years of birth history data from three rounds of the National Family Health Survey (NFHS). The random effects dynamic probit model was used to decompose the decline in infant deaths into the contributions by the socio-economic and demographic factors, including the lagged independent variable, the previous infant death measuring the clustering of infant deaths in families. The study found that there has been a decline in the clustering of infant deaths among families during the past two and half decades. The simulation result shows that if the clustering of infant deaths in families in India was completely removed, there would be a decline of nearly 30 percent in the infant mortality rate (IMR). A decomposition analysis based on the dynamic probit model shows that for NFHS-1 and NFHS-3, in the total change of the probability of infant deaths, the rate of change for a given population composition contributed around 45 percent, and about 44 percent was explained by a compositional shift. Between NFHS-3 and NFHS-4, the rate of change for a given population composition contributed 86%, and the population composition for a given rate contributed 10% to the total change in the probability of infant deaths. Within this rate, the contribution of a previous infant was 0.8% and the mother’s age was 10%; nearly 31% was contributed by the region of residence, 69% by the mother’s education, and around 20% was contributed by the wealth index and around 8.7% by the sex of the child. The mother’s unobserved factors contributed more than 50 percent to the variability of infant deaths in all the survey rounds and was also statistically significant (*p* < 0.01). Bivariate analysis suggests that women with two or more infant losses were much less likely to have full immunization (10%) than women with no infant loss (62%), although institutional delivery was high among both groups of women.

## 1. Introduction

One of the 13 specific targets for the Sustainable Development Goal 3 included, the end, by 2030, of preventable deaths of children under five years of age, with all countries aiming to reduce under-five mortality to a level at least as low as 25 per 1000 live births. The meeting of this target requires a significant reduction in infant mortality in India, and the challenge has been to understand the underlying determinants of infant deaths in India. Many studies of different disciplines and periods have indicated that the infant/child mortality problem in India is complex. The possible explanations include economic growth, poor hygiene, anaemia before and during pregnancy, infections, hunger, and epidemics [1,2,3].

In the past few decades, India has made a significant reduction in the infant mortality rate (IMR), from 88 infant deaths per 1000 live births in 1990 to 37 infant deaths per 1000 live births in 2015 [4]. It has been found that apart from the known risk factors affecting infant mortality, there is a tendency of infant deaths to cluster among smaller numbers of mothers/families [5,6,7,8]. It implies a heterogeneity in the risk of experiencing infant deaths; that is, a few mothers are more susceptible to experience child deaths than other women. This is known in demographic literature as death clustering. Death clustering has been defined consistently in the following ways. Firstly, it has been defined as the counting of the number of women who have experienced more than one child loss and the extent of the deaths concentrated in such families [7,9,10,11]. Many studies viewed death clustering as being a greater heterogeneity in the distribution of child deaths across families than would be expected if the deaths were distributed randomly [12,13,14,15]. In addition to the above definitions, death clustering has been viewed as what is left unexplained after the observed correlates are controlled, and it is thus attributed to unobserved or unobservable genetic, behavioral, and environmental factors related to mortality [5,6,10,12,13,16,17,18,19,20,21,22].

Death clustering has two important dimensions in the analysis of mortality. First, siblings, due to their shared familial, genetic, and socio-economic environment, make the survival status of the children in some families riskier than in other families. Thus, in the clustered data, modeling infant deaths does not assume an independence of the observations, which would lead to violations of the assumptions of the regression model. Second, the unexplained heterogeneity (a measure for interfamily variation) assumes the variation in the level of mortality risk among families [9,12]. In India, observed factors such as income disparities, uneven regional development levels, the mother’s educational status, caste, religion, age of the mother, etc., are known to play a major role in affecting infant mortality [23,24,25,26,27,28], but the death of a previous child in families along with the unobserved residual variation is found to be minimally addressed. 

Guo (1993) has shown, in Guatemala, that household income and the mother’s educational attainment are two of the most important determinants of death clustering at the familial level [10]. Fuchs, Pamuk, and Lutz (2010), using the Demographic and Health Survey (DHS) data for developing countries, assessed the relative role of education and wealth in reducing child mortality and found that in virtually all of the models the mother’s education mattered more for infant survival than household wealth [29]. Garenne and Garenne (2003) used the wealth index as a discriminatory tool for screening families at a higher risk of infant and child mortality [30]. Sastry (1997), when comparing the results of the standard hazards model with those of the hazards model with single random effects for each family (or frailty model), found that there was an increase in the absolute magnitude of the coefficients for the maternal education (14 percent) and household income (10 percent), respectively [16]. Ranjan, Dwivedi et al. (2018) found a host of factors affecting the clustering of infant deaths in different states of India and among different caste groups in the central and eastern regions of India [26]. Arulampalam and Bhalotra (2008) identified the effect of the clustering of infant deaths after discounting the mother and found a level of unobserved factors in 13 out of 15 states of India, representing different regions of the country [6]. 

In the present paper, we intend to examine the changing nature and patterns of the familial clustering of infant deaths in India and for some of the selected bigger states between two surveys, the National Family Health Survey (NFHS-1) (1992–1993) and NFHS-4 (2015–2016). This paper further presents various optimistic scenarios of gain in the reduction in IMR by reducing infant death clustering among families in India. Finally, we used the regression decomposition technique to find out the contribution of various factors, including the clustering of infant deaths (as measured through the previous death coefficient in the model), the regional differentials in infant mortality, the wealth index, and the mother’s education, in explaining the decline in infant mortality in India over time.

## 2. Materials and Methods

The data for the study were taken from the National Family Health Survey based on large representative samples in India. The birth history data from the three survey rounds of the National Family Health Survey conducted, namely NFHS-1 in the year 1992–1993, NFHS-3 in the year 2005–2006, and NFHS-4 2015–2016, have been utilized for the study. This dataset has one record for every child ever born to the interviewed women. Essentially, it is the full birth history of all the women interviewed, including information on pregnancy and postnatal care as well as immunization and health for the children born in the last 5 years. The data for the mother of each of these children are also included. This file can be used to calculate health indicators as well as the fertility and mortality rates. The unit of analysis (case) in this file is all the children that were ever born to the eligible women. In NFHS-1, there was an overall sample size of 89,777 women, who were or had been married and were in the age group of 13 to 49 years; their total births amounted to 275,172 between 1954 and 1993 (nearly forty years). NFHS-3 captures the information of about 256,782 births by 124,385 women aged 15–49 years between 1968 and 2006 (nearly 38 years). NFHS-4 has the information of about 1,315,617 births by 699,686 mothers in the age group of 15 to 49 years between 1970 and 2016 (46 years). In all three survey rounds, the information related to all the births, such as year of birth, birth order, sex of the child, current age of the child, etc., along with children’s survival status and age of death, is considered for a period spanning nearly 35 years in order to examine the family-level death clustering. 

All the analyses were based on these retrospective birth histories, but for the purpose of the multivariate and decomposition analysis, we took truncated 10 years of the birth histories of NFHS-1, NFHS-3, and NFHS-4. The rationale behind utilizing 10 years of birth history relates to the need to include the household wealth variable in our analysis. To examine the maternal competence factor in utilizing the health care services, the behavior of mothers with respect to their children was analyzed using Kids File. The Kids File dataset has one record for every child of the interviewed women born in the five years preceding the survey. It contains the information related to the child’s pregnancy and postnatal care and to immunization and health. The data for the mothers of each of these children are included. The unit of analysis (case) in this file is the children of women born in the last 5 years (0–59 months). In order to capture the difference in behavior between disadvantaged group of mothers and the relatively better off mothers in terms of experiencing infant deaths, we have defined low-risk and high-risk mothers. The low-risk mothers’ group is defined as being those mothers who experienced either no infant death or one infant death, while the high-risk mothers are those mothers who have experienced multiple child loss (two or more child deaths).

The family-level extent of the infant death clustering was examined by using the bivariate analysis of all the children ever born to the mothers (or families) and the number of infant deaths experienced by the mothers. The intra-class correlation coefficient (ICC) and the median odds ratio (MOR) were estimated through a multilevel random effect in a logit model [31]. 

Simulation analyses were carried out by randomly altering the dataset directly so that all the mothers/families with two or more infant deaths (i.e., clustered deaths) altered to experience with exactly two infant deaths were in situation 1; again all the mothers/families with two or more infant deaths (i.e., clustered deaths) altered to experience with exactly one infant death were in situation 2; and finally all the mothers/families with two or more infant deaths (i.e., clustered deaths) altered to experience with exactly no infant deaths were in situation 3. In each of these three situations with the altered dataset, the mortality was estimated using the DHS methodology of the direct estimation of child mortality from birth histories. Decomposition analysis was performed on random effects dynamic probit model.

### 2.1. The Statistical Model: Random Effects Dynamic Probit Model

The model used here is the random effects dynamic probit model. The model controls the socio-economic and demographic factors along with an independent lagged variable, the previous infant deaths whose coefficient will capture the impact of the clustering of infant deaths in families. The previous infant death is correlated with the error term; however, the present model deals with this endogeneity problem by modelling on the first child. The description for the model is shown below. 

Let there be $n_{i}$ children in the *i*th family. For child *j* (*j* = 2, …, $\;n_{i}$) in family *i* (*i* = 1, 2, …, *N*), the unobservable propensity to experience an infant death, $Y_{ij}^{*}$, is given by:(1)$$Y_{ij}^{*}=X_{ij}^{\prime }\beta +\gamma \;Y_{ij-1}+\alpha_{i}+u_{ij}$$
where *X* is a vector of strictly exogenous child and family specific characteristics. *β* is the vector of coefficients associated with *X*. An infant is observed to die when his or her propensity for death crosses a threshold; in this case, this is when $Y_{ij}^{*}$ > 0, and the binary outcome is denoted as $Y_{ij}^{*}$ = 1. The term $\alpha_{i}$ captures unobserved heterogeneity. It accounts for all the time-invariant unobserved family characteristics, and it will also include genetic characteristics and should be interpreted as the ‘average’ scarring effect (state of dependence or causal effect of previous infant death on the survival status of the next child) over the time.

The joint probability of the sequence of binary outcomes depends upon α. Hence, we need to define some initial conditions for P($Y_{1}|\;\alpha )$. If there was no unobserved heterogeneity α, the initial condition *Y*_1_ could be treated as exogenous and the model in (1) could be estimated using the sample of children (*j* = 2, …, *n*). Thus, the mortality risk of the first-born child of each mother is:(2)$$Y_{i1}^{*}=Z_{i}^{\prime }\omega +\theta \;\alpha_{i}+u_{i1}\;i=1,\ldots ,N\;\;\;$$
where $\;Z_{i}$ is a vector of the exogenous covariates; the vector of covariates in *X* and Z need not be the same and need not equal one. Equations (1) and (2) together specify the complete model for the infant survival process.

The estimation procedure of the random effects dynamic probit model and its detailed description can be found elsewhere in the papers [5,6]. In addition, the random effects dynamic probit model was previously applied in other studies of death clustering [14,19,32,33].

As with the decomposition of other regression models, the decomposition of the random effects dynamic probit model also yields three components, namely the rate part for a given population composition; the composition part for a given rate; and the interaction part, which deals with the joint effect due to the interaction between the rate and the composition. 

### 2.2. Decomposition of the Random Effects Dynamic Probit Model

If *x_k_* (*k* = 1, 2, …, *l*) is the “*l*” mother and child-specific covariates in the model, then the difference is $\left[\;{y_{ij}^{*}}_{NFHS-4}-{y_{ij}^{*}}_{NFHS-3}\;\right]$, or it is decomposed using the following equation: $$\left[\;{y_{ij}^{*}}_{NFHS-4}-{y_{ij}^{*}}_{NFHS-3}\;\right]=\left(\alpha_{i\left(4\right)}-\alpha_{i\left(3\right)}\right)+\overset{l}{\underset{k=1}{\sum }}P_{k\left(3\right)}*\left(\beta_{k\left(4\right)}-\beta_{k\left(3\right)}\right)+\;\sum_{k=1}^{l}\beta_{k\left(3\right)}*\left(P_{k\left(4\right)}-P_{k\left(3\right)}\right)+\sum_{k=1}^{l}\left(\left(P_{k\left(4\right)}-P_{k\left(3\right)}\right)*(\beta_{k\left(4\right)}-\beta_{k\left(3\right)}\right))+P_{\left(3\right)}*\left(\gamma_{\left(4\right)}-\gamma_{\left(3\right)}\right)+\gamma_{3}*\left(P_{\left(4\right)}-P_{\left(3\right)}\right)+\left(\gamma_{\left(4\right)}-\gamma_{\left(3\right)}\right)*\left(P_{\left(4\right)}-P_{\left(3\right)}\right)$$
where

$P_{k\left(m\right)}$ = The proportion of the *n*th level of the *k*th covariate in the *m*th NFHS survey.

$P_{\left(m\right)}$ = The proportion of the preceding sibling death in the family in the *m*th NFHS survey.

$\alpha_{i\left(m\right)}$ = The *i*th mother-level unobserved heterogeneity in the *m*th NFHS survey.

$\gamma_{\left(m\right)}$ = The clustering coefficient in the *m*th NFHS survey.

$\beta_{k\left(m\right)}$ = The coefficient of the *n*th level of the kth covariate in the *m*th NFHS survey.

*m* = 3, 4. 

(3) denotes NFHS-3 and (4) denotes NFHS-4.

This procedure yields three components: the composition, the rate, and the interaction components. 

The dependent and independent variables included in the model are defined below.

### 2.3. Empirical Model

#### 2.3.1. Dependent Variable

The survival status of the index child and the previous child’s survival status are defined as “1” if the child died as an infant and “0” otherwise. Only singleton births were taken for both survey rounds. We dropped women who were never married from NFHS-3 (2005–2006) and NFHS-4 (2015–2016). We also dropped from both surveys those children whose age was less than 12 months at the time of survey as these children did not have one full year of exposure. We did our all analysis in STATA 13.1.

#### 2.3.2. Choice of Covariates

Our choice for the covariates for possible inclusion in our model was governed by both by the theoretical background and those covariates that showed at least some significant relationship in case of bivariate analysis. Wealth (poorest/poorer/middle/richer/richest); mother’s age (15–29 years/30–39 years/40–49 years); region, divided into six categories (central/north/east/northeast/west/south); place of residence (rural/urban); mother’s education (illiterate/primary/secondary/higher); religion (Hindu/others); caste (other caste/SC and ST); child sex (male/female); and birth order (2/3 to 5/5+) were included in our model.

## 3. Results 

Table 1 and Table 2: shows the levels of infant death clustering in families and its changes in India and its selected states for two and a half decades. The results showed that in NFHS-1, infant mortality in India was 79 infant deaths per 1000 live births; of the total infant deaths, more than half of the deaths were concentrated in nearly seven percent of the families. The states with more than half the total deaths as clustered deaths are Rajasthan, Uttar Pradesh, Bihar, Assam, Jharkhand, Odisha, Chhattisgarh, and Madhya Pradesh. The worst among them is Uttar Pradesh where a level of nearly 62 percent clustered infant deaths was experienced by 14 percent of the families. 

Even after a gap of one and half decades from NFHS-1, in NFHS-3, 46 percent of the infant deaths were concentrated in just 5 percent of the families in India, and 1.3 percent of the families experienced three or more infant deaths, which contributed to nearly 18 percent of the total infant deaths in the sample. Only three states, namely Rajasthan, Uttar Pradesh and Madhya Pradesh, have more than half the total infant deaths as clustered deaths. 

In NFHS-4 (2015–2016), in India, there were 41 infant deaths per 1000 live births, and two percent of the families experienced two or more infant deaths and contributed nearly 37 percent of the total infant deaths. Furthermore, less than one percent of the families in India had experienced three or more infant deaths and contributed nearly 13 percent of the total infant deaths. So, most of the clustered infant deaths experienced by the families comprised two infant deaths. States such as Uttar Pradesh, Madhya Pradesh, and Bihar have almost 5 percent, 3 percent, and 3 percent of the families, respectively, who had experienced multiple infant deaths, and the extent of the clustered deaths in such families was more than 40 percent. 

By examining the relative change of the clustering of infant deaths in families in India, there was a reduction of 34 percentage points between NFHS-1 and NFHS-3 and 57 percentage points between NFHS-3 and NFHS-4 in families with two or more infant deaths, respectively. Between NFHS-1 and NFHS-3, all the other states, except for Rajasthan, experienced a reduction in the percentage of families with two or more infant deaths as well as a reduction in the percentage of clustered deaths. However, between NFHS-3 and NFHS-4, Kerala experienced no change in the percentage of clustered infant deaths, although the percentage of families experiencing clustered infant deaths was reduced by 66 percent. In all the states, the extent of the reduction in some families was in having two or more infant deaths. 

Table 3 presents the percentage of families with two or more infant deaths (high-risk families) and the level of death clustering according to selected background variables in NFHS-1 (1992–1993), NFHS-3 (2005–2006), and NFHS-4 (2015–2016), along with their relative change.

Between NFHS-1 and NFHS-3, for the mothers belonging to scheduled tribes, there has been almost no relative change in the percentage of families with clustered deaths and the extent of the deaths clustered in them. The relative change between NFHS-3 and NFHS-4 in India shows that, for the mothers aged 30 years or more and for the mothers who received higher education, both the percentage of families with two or more infant deaths and the percentage of clustered infant deaths in them has increased. In all the subgroups of the different characteristics both the percent of families with clustered deaths and the extent of the deaths clustered in them was reduced between NFHS-1 and NFHS-3 and also between NFHS-3 and NFHS-4.

Based on the simulation analysis, the estimated mortality in the three different situations of the clustering of infant deaths in families is presented in Table 4. In NFHS-3, the results of the simulation analysis show that at the India level, a reduction of 37 percent in IMR could be achieved if we remove completely the clustering of deaths in families in the country (column 7a). Once the death clustering in families has been removed completely, the burden of infant deaths could be reduced by more than 40 percent in the states of Uttarakhand, Rajasthan, Uttar Pradesh, Jharkhand, Chhattisgarh, and Madhya Pradesh. The NFHS-4 data reveal that eliminating clustering completely from families in India would help in reducing infant mortality by 27 percent and nearly a one-third reduction in infant mortality could be achieved in the states of Uttar Pradesh, Uttarakhand, Bihar, Assam, and Madhya Pradesh. This shows an opportunity to achieve the sustainable development goal related to goal 3 and to target 3.2 to end the preventable deaths of newborns and children under 5 years of age. 

Table 5 presents the intra-class correlations (ICCs) and median odds ratios (MORs) for India and the selected states in NFHS-1, NFHS-3, and NFHS-4. The null model presents the results of the ICCs and MORs in the case without consideration of any covariates, whereas model 1 presents the ICCs and MORs when the previous death in the families was included in the regression analysis.

For Model 1, between NFHS-1 and NFHS-3, there was a 7 percent increase in ICC, and between NFHS-3 and NFHS-4, the increase in ICC was 39 percent. This indicates that over time the clustering among families in India has increased and that it increased more rapidly between NFHS-3 and NFHS-4. In fact, all the states marked the positive increase in ICC between NFHS-3 and NFHS-4, with states such as Rajasthan, Uttar Pradesh, Jharkhand, and Gujarat indicating more clustering of infant deaths among families. Even the MORs have been positive during these two survey rounds for India, indicating that high-risk mothers further intensified the risk of infant deaths.

Table 6 shows the results of the dynamic probit model and its decomposition analysis for the NFHS-1 and NFHS-3 period. For NFHS-1 and NFHS-3, the coefficients in the dynamic probit model for the factors such as previous infant deaths, region of residence, mother’s education, and wealth index were found to be statistically significant (*p* < 0.01). In NFHS-1, the 38 percent (0.6212/0.6212 + 1) variation in the risk of infant death is due to unexplained mother-level unobserved factors. In NFHS-3, the estimated mother-level unobserved heterogeneity reduced from 0.6212 to 0.514, and it was statistically significant. The decomposition of the overall change in infant deaths resulted in its components, which are propensity (for a given population), composition (with a given rate), and interaction term (joint effect), and the scores of the composition and interaction terms at the aggregate and sub-group level. It is evident from the table that in one and a half decades, the foremost component of the decrease in the probability of infant death was the change in the propensity for experiencing infant deaths, which explains around 45 percent of the total absolute change. About 44 percent of the total absolute change can be explained by compositional shift of the population. The signs for the propensity factors were positive in the case of previous infant death, region of residence, place of residence, religion, caste, and wealth index, while they were negative for the mother’s age, the mother’s education, and the sex of the child. The positive propensities indicate that the rate of occurrence of infant deaths among subgroups with respect to the omitted category has increased while the negative propensities contribute to the proposition that the direction of the change in the rates of occurrence of infant deaths has narrowed down among different sub-groups with respect to the reference category. The statistically significant factors which contributed to the total change in infant deaths are around 1.7 percent (14.5/379.4 + 368.9 + 89.5), which is contributed by previous infant death; 26 percent is contributed by region of residence, 8 percent is contributed by place of residence, 31 percent is contributed by mother’s education, and 36 percent is contributed by the wealth index, respectively. The comparison of the two probit models also reveals a negative increase in the intercept, indicating that even when all the explanatory variables are set to be equal to their reference categories, the probability of infant deaths was lower in NFHS-3 than in NFHS-1.

Almost all of the decrease in infant deaths that took place due to shifts in the population structure for a given rate of occurrence of infant deaths was explained mainly by the mother’s education and the occurrence of previous infant deaths in the families. These two are also the statistically significant factors. Due to the change in population composition for a given rate, the women’s education explained around 35 percent, and previous infant deaths in the families contributed around 3 percent to the overall decrease in infant deaths. 

The interaction term contributed nearly 10 percent (89.5/(379.4 + 368.9 + 89.5)) to the overall decline in infant deaths between 1992 and 1993 and between 2005 and 2006.

Table 7 shows the results of the dynamic probit model and its decomposition analysis for the NFHS-3 and NFHS-4 period. It is evident from the table that rate of occurrence of infant deaths for a given population composition contributes 86 percent to the overall absolute change in the decreasing of the probability of infant death in the NFHS-3 and NFHS-4 period. The compositional shift for a given rate contributed nearly 8 percent to the decline in infant deaths. Within this rate, the signs of the propensity factors were positive for previous infant death, region of residence, place of residence, mother’s education, religion, and sex of the child, which indicates that the rate of infant deaths among the subgroups with respect to the omitted category has increased. Signs of negative propensity were found in the case of the mother’s age, caste, and wealth index, which means the direction of the change in rates shows a declining trend among different sub-groups with respect to the reference category. Within the rate for a given population composition, of the total absolute change of infant deaths the contribution of a previous infant was found to be 0.8%, the mother’s age contribution was 10%; nearly 31% was contributed by region of residence, 69% by mother’s education, and around 20% by the wealth index and around 8.7% by the sex of the child.

Table 8 and Table 9 show the socio-demographic characteristics and health care utilization patterns of the mothers experiencing no child loss, one child loss, and the loss of two or more children in the three survey rounds of NFHS-1, NFSH-3, and NFSH-4. In NFHS-1, of the mothers who experienced the loss of two or more children, 12 percent of them attended at least four ANC checkups; thirty percent of the mothers had consumed IFA tablets during pregnancy; nearly 30 percent had taken two or more tetanus injections during pregnancy, while 28 percent of the mothers with no child loss went for four or more ANC checkups; more than 50 percent of the mothers had taken an IFA tablet, and the same percentage of women had taken two or more tetanus injections during pregnancy. There was not much difference in the institutional delivery between all three groups of mothers. However, there was a huge gap in giving full immunization to their children between the mothers with multiple child loss and the mothers with no child loss. In NFHS-4, more than half of children who had mothers with no child loss went for ANC, while 30 percent of the mothers with multiple infant deaths went for ANC checkup. Those who consumed IFA tablets received two or more TT injections, and the percentages of institutional deliveries were similar for the three groups of women. By examining the full immunization, just 10 percent of the children of the mothers with multiple deaths received full immunization, while more than 60 percent of the children whose mothers had not had any child loss were fully immunized.

## 4. Conclusions

This study assesses the determinants of infant mortality and the clustering of deaths in India and in some major states. The results of this study suggest that infant mortality and the clustering of deaths in families in India declined between NFHS-1 (1992–1993) and NFHS-4 (2015–2016), though the pace of reduction with regard to the clustered deaths within families is much slower than the reduction in the high-risk families for both time periods between NFHS-1 and NFHS-3 and between NFHS-3 and NFHS-4. The tempo of the reduction in clustered infant deaths in families and the reduction in high-risk families was much faster between NFHS-3 and NFSH-4.

During the interval between NFHS-3 and NFHS-4, a significant development was the launch of the National Rural Health Mission (NRHM) in April 2005, a flagship program by the Government of India to tackle the high burden of maternal, neonatal, and infant mortality among India’s rural populations. More recently, the Government of India (GOI) named the program the National Health Mission (NHM) to also include coverage of the urban poor. Through the NHM, the GOI initiated programs such as the Janani Suraksha Yojana (JSY) in 2005, which provides conditional cash transfers to incentivize women to give birth in a health facility rather than at home. Key aspects of the NHM are its enormous scale, its focus on extending services to the poor, and its inherent flexibility for introducing innovative approaches and for improving health system responses to improve reproductive, maternal, newborn and child health. The policy is instrumental in promoting newborn care, proper counselling, and widespread messages on proper breastfeeding practices and food supplementation at the right time and a complete package of immunization for children. Janani Suraksha Yojana was incentive-based institutional delivery, and it helped a lot in tackling child mortality as well as maternal deaths in families. Even toll-free emergency helpline numbers, such as 112, 102, and 108, for calling ambulances and other emergency health services in even the remotest part of the country helped to reduce critical newborn outcomes. With the exception of Rajasthan, the pattern of the other states followed the national pattern of the reduction in infant death clustering in families in India. In Rajasthan, high-risk families, and the proportion of clustered infant deaths among these families increased between NFHS-1 and NFHS-3, with some reduction during the NFHS-3 and NFHS-4 period. This suggests that among the high-mortality states such as Rajasthan, the clustering of infant deaths in few families is persisting, and the health-related policies and programs, though they have received attention, were not that effective, and this needs to be reframed.

The bivariate percentage distribution of the relative change of the clustering of infant deaths in families between NFHS-3 and NFSH-4 highlights an increased clustering for women with an age greater than 30 years and among the higher-educated groups of women. These peculiar findings are contrary to the prior findings because a mother’s education and her experience in child rearing had a negative association with the clustering of infant deaths in families [7]. This may be because India underwent many socio-economic transformations during that period, which transformed the lives of many women, including women in higher as well as lower socio-economic groups. There was also an upsurge in employment among women of the middle- and higher-income groups, which probably resulted in lesser attention being paid to children’s health. According to the Uttar Pradesh Technical Support Unit (UP TSU) report, the evidence shows that institutional deliveries among younger and low-parity women have increased significantly, especially in the areas where infant mortality was much higher than the national average [34]. This is true irrespective of education. This could be the result of newly introduced successful programs, such as NRHM and Janani Suraksha Yojana, which are utilized by younger women irrespective of their education. This was possible due to the work of frontline workers such as ASHAs and the progressive attitude of the younger generation. 

The findings based on the NFHS-3 and NFSH-4 data suggest that, in India, if the clustering of infant deaths in families was removed completely, there would be, roughly, a decline of nearly 30 percent in IMR. A reduction in IMR by 30 percent can be achieved just by identifying the areas where these high-risk families are located and employing targeted interventions. As indicated by the intra-class correlation (ICC) model, except for a few states, the families of most of the states experienced an increase in the number of multiple infant deaths between NFHS-1 and NFHS-3, while the situation worsened further between NFHS-3 and NFHS-4 as per the ICCs. This highlights the importance of the clustering of infant deaths in families. This suggests that families which are disadvantaged are continuing to be disadvantaged. For instance, the people of the scheduled caste are prime examples of this scenario, and the results support this argument. The findings from the decomposition analysis of the change in infant mortality between NFHS-1 and NFHS-3 suggest that the overall rate of change in infant mortality for a given population composition played a more important role than the change in composition for a given rate. The major contributors within the rate which caused the infant mortality to decline in the country between 1992 and 1993 and between 2005 and 2006 were mainly the wealth index, the mother’s education, the region of residence, and previous infant death. Regarding the mother’s education, illiterate and lower-educated mothers have a higher rate of decline in infant mortality than the higher-educated mothers. It may be possible that more educated mothers have already achieved lower mortality, but other subgroups of education due to their disadvantageous position in the society have had higher mortality. Over the years, due to change in their behavior, they attained maximum gain in reducing the mortality. This also suggests that the government policies which focus on the disadvantaged groups might be partly responsible for this. The change in the propensity of the region covariate also contributed largely to reducing the probability of infant deaths. The probability of infant deaths in different regions of the country increased with respect to the central region in the country. The other factors may be economic and social; these factors created a lot of regional imbalances in the use of the health infrastructure in the country. The small increase in the propensity of previous infant death factors shows a clear increase in the clustering effect in families between 1992 and 2006. 

Similar arguments were made with the respect to the important role socio-economic development plays in mortality transition. The decline in infant/child mortality has been cited as the most important and significant factor in demographic transition [35]. Others have highlighted the role of female education [36,37]. In addition to socio-economic conditions and child survival, the role of fertility transition is also recognized [38]. Clearly, while the importance of socio-economic changes was recognized, as emphasized in the Bucharest World Conference, it was found that there was a strong correlation between fertility and infant mortality, especially neo-natal mortality. For instance, the north-central region of India, with low socio-economic development, experienced higher fertility and higher infant and neo-natal mortality. To complicate it further, it was also found that there was a high correlation between maternal mortality and neo-natal mortality (a major contributor to infant mortality). The maternal and neo-natal deaths can be prevented by preventing the delays in deciding to seek care, reaching the health care delivery institutions, and receiving adequate and emergency care in these facilities [39,40]. To address these delays and promote the timely use of skilled maternal and neonatal care, the WHO and other international agencies recommended the strategy of birth preparedness, which involves preparation for childbirth during pregnancy, including identifying the location of the facility for birth, ensuring funds for birth-related expenses, and identifying the mode of transport for reaching the identified facility, among others [41,42]. 

One of the important drivers of infant and neo-natal mortality is the health status of the newborn. A recent study in India on maternal nutritional status found that women suffering from anaemia, especially during pregnancy, are at a higher risk of poor birth outcomes, such as preterm birth and low birth weight, due to weak intrauterine growth [43]. There is evidence to show that half of the expectant mothers, children, and adolescent girls in India suffer from anaemia [44]. It was also found that half of the burden of anaemia is assumed to be due to iron deficiency, and both folic acid and iron deficiency during pregnancy are important factors for preterm delivery, anaemia, low birth weight, and, in turn, increased stunting among children [45,46]. The World Health Organization suggested that those pregnant women who attend ANC meetings should be given a recommended dose of 30–60 mg iron and 400 mg folic acid [44,47]. In India, IFA consumption has been low among pregnant women despite the fact that IFA supplements are distributed free, especially during ANC visits. This may be partly because full (at least four or more visits) ANC coverage nationally was only 59% during their last pregnancy [48]. The policymakers also think that that there is a problem of adherence. This is not entirely true as there seems to be a fundamental health system problem. According to one study, there is a serious problem of stock and an untimely supply of IFA supplements in government facilities rather than poor adherence to the supplement by the women, which has been the major concern of policymakers [48]. Among those women who attended one or two ANCs, less than 50% of the pregnant women received the recommended 100 or more IFA supplements [48]. Other factors which are equally important in improving the health of the newborn are the initiation of breast feeding in the first hour of birth, exclusive breast feeding during the first six months of the baby, and kangaroo mother care.

The period between NFHS-3 and NFHS-4 was marked by a maximum contribution to the decline followed by a compositional change of population. The overall rate in declining infant deaths between NFHS-3 and NFSH-4 is contributed to mainly by the mother’s education, the region of residence, and the wealth index, and the contribution was maximum by the region of residence. It shows that in comparison with the central region of the country every other region experienced a decrease in infant deaths, and it was maximum in western India, reflecting the socio-economic development. Similarly, in India during this time, compared to higher-educated women, illiterate women experienced greater decrease in the rate of experiencing infant deaths. A few researchers emphasized the role of globalization and associated economic gain rather than scientific medicine as being that which has led to the decline in mortality [1,2,49]. However, other researchers argued that the path of decline in child mortality is least correlated with the mother’s education [50]. Later, Caldwell found a more complex dependency between state of health and cultural, social, and lifestyle factors, especially with regard to child mortality, than is commonly believed [51,52]. In his highly influential article, Caldwell explored the social and political pathways to mortality success, especially among countries with high mortality which led new pathways, particularly with regard to the reduced importance of female education as a determinant of child mortality. Caldwell highlights how interactions between social consensus, health care systems, and human capital dependence offer a pathway to reduced mortality. This supports in some way the Caldwell, McKeown, and Omran historical perspectives on the decline of mortality in India. Few recent studies believe in geographic determinism, that is, the location of the countries, developed and developing countries, being in certain zones with environmental conditions conducive to the decline in mortality [53].

The comparison of the characteristics of both high-risk women and low-risk women in terms of basic socio-demographic characteristics, childcare practices, hygiene practices, mass media exposure, and utilization of pre- and post-natal services mattered in all the three survey rounds. It has been observed that the main difference between these groups of women exists in education, parity, utilization of ANC, and full immunization.

## 5. Implications

The study shows a clustering of infant deaths among some disadvantaged women in the country. Previous studies explored the clustering of infant deaths in families and demonstrated that part of the clustering was due to socio-economic and demographic factors. The present study, however, shows that education, age, and wealth quantiles do not completely corroborate the earlier findings, and therefore, further studies are needed to explore and verify the role of education, age, and wealth quantiles.

The findings of the study are expected to be useful for interventions when working with high-risk women against the clustering of infant deaths. If some women are experiencing multiple infant deaths, it is imperative to know the key factors that affect this phenomenon and work towards addressing such issues effectively through counseling and directing them towards appropriate health care for high-risk pregnancies. For example, for disadvantaged women, the specific interventions should be the focus for such communities, and frontline workers can create an enabling environment for positive pregnancy outcomes. More importantly, the knowledge of clustering would help policy makers and program implementers to focus on high-risk women.

## Figures and Tables

**Table 1 ijerph-19-14384-t001:** Levels of clustering of infant deaths in families in India and its selected states, 1992–2016.

	NFHS-1 (1992–1993)							NFHS-3 (2005–2006)							NFHS-4 (2015–2016)						
States	(1)	(2)	(3)	(4)	(5)	(6)	(7)	(1)	(2)	(3)	(4)	(5)	(6)	(7)	(1)	(2)	(3)	(4)	(5)	(6)	(7)
India	7.3	52.3	2.39	24.5	79,350	24,976	79	4.8	45.9	1.3	18.2	84,609	17,796	57	2.06	36.8	0.5	12.94	476,619	66,158	41
Uttarakhand	5.6	47.0	1.8	23.0	1549	534	62	3.7	44.5	0.8	14.4	1985	385	41	2.0	35.1	0.5	12.38	11,440	1568	40
Rajasthan	5.5	51.4	1.8	24.1	4497	1237	77	7.2	52.4	2.1	23.0	2821	957	66	2.1	34.6	0.5	11.6	28,874	4133	41
Uttar Pradesh	13.5	61.7	5.4	34.0	8491	5058	98	9.2	53.0	2.9	23.0	8451	3302	73	4.5	43.6	1.3	17.53	61,898	15,714	64
Bihar	9.1	53.1	2.6	22.2	3973	1594	84	6.1	46.1	1.7	18.5	2743	850	63	3.1	40.0	0.8	14.7	32,507	6036	48
Assam	8.5	51.0	2.4	21.2	2717	1006	91	4.4	46.3	1.1	18.9	2565	583	67	2.2	38.5	0.5	13.1	19,922	2631	48
West Bengal	6.2	46.9	1.8	20.2	3782	1289	65	3.1	38.7	0.8	14.4	4792	832	48	1.1	26.4	0.1	4.76	13,146	1319	27
Jharkhand	6.9	52.9	2.1	23.3	1112	340	70	5.8	44.6	1.4	15.6	2134	611	69	2.4	35.2	0.5	11.1	20,253	3096	44
Odisha	10.8	54.6	3.6	25.5	3782	1793	101	5.6	44.9	1.6	18.8	3101	887	63	2.5	37.6	0.6	13.43	22,924	3793	40
Chhattisgarh	8.1	54.5	2.1	21.3	1022	354	62	6.5	47.4	2.0	20.9	2638	842	72	2.8	36.3	0.7	14.0	16,660	2993	54
Madhya Pradesh	9.2	57.4	3.2	28.7	4369	1845	89	8.2	54.7	2.5	23.8	4669	1420	67	3.3	41.7	0.9	15.9	44,295	8285	51
Gujarat	5.3	43.9	1.4	17.0	3390	958	62	3.8	41.0	0.7	11.7	2654	565	51	1.7	36.1	0.4	12.3	16,123	1844	34
Maharashtra	4.41	45.9	1.3	20.3	3673	883	47	2.1	36.7	0.6	14.1	6174	855	36	0.7	25.3	0.2	7.2	21,042	1493	24
Kerala	1.6	31.5	0.3	9.9	3896	454	28	0.8	29.3	0.2	8.4	2479	143	15	0.3	29.4	0.1	8.8	7660	179	6
Tamil Nadu	4.8	43.4	1.3	17.3	3502	950	65	2.0	31.9	0.4	10.4	4154	562	32	0.6	26.7	0.2	8.97	20,582	1137	21

Note: (1) Families with 2 or more infant deaths; (2) 2 or more infant deaths clustered; (3) families with 3 or more deaths experienced; (4) three or more infant deaths; (5) total families experiencing infant death (number); (6) infant deaths number; (7) infant mortality rate (IMR).

**Table 2 ijerph-19-14384-t002:** Relative change in clustering of deaths in India and selected states, 1992–2016.

States	Relative Change NFHS-3 to NFHS-4			Relative Change NFHS-1 to NFHS-3		
	Two or More Infant Deaths (IDs) Clustered	Families with 2 or More IDs	IMR	Two or More Infant Deaths (IDs) Clustered	Families with 2 or More IDs	IMR
India	−20	−57	−28	−12	−34	−27
Uttarakhand	−21	−45	−2	−5	−34	−34
Rajasthan	−34	−71	−38	2	31	−15
Uttar Pradesh	−18	−51	−12	−14	−32	−26
Bihar	−13	−49	−23	−13	−33	−26
Assam	−17	−50	−28	−9	−48	−26
West Bengal	−32	−64	−43	−17	−50	−27
Jharkhand	−21	−59	−36	−16	−16	−2
Odisha	−16	−55	−36	−18	−48	−38
Chhattisgarh	−23	−57	−25	−13	−20	15
Madhya Pradesh	−24	−60	−24	−5	−11	−24
Gujarat	−12	−54	−33	−7	−28	−19
Maharashtra	−31	−66	−33	−20	−52	−24
Kerala	0	−66	−60	−7	−50	−45
Tamil Nadu	−16	−69	−33	−26	−58	−52

**Table 3 ijerph-19-14384-t003:** Infant death clustering in families and extent of clustered deaths in families by selected background characteristics, India, 1992–2016.

Background Characteristics	NFHS-1 (1992–1993)		NFHS-3 (2005–2006)		NFHS-4 (2015–2016)		Relative Change NFHS-1 to NFHS-3		Relative Change NFHS-3 to NFHS-4	
	Families	Two or More Deaths	Families	Two or More Deaths	Families	Two or More Deaths				
	(1)	(2)	(3)	(4)	(5)	(6)	(7)		(8)	
Age at First Birth										
≤20 yrs	7.6	52.7	5.0	46.2	2.6	39.1	−34	−12	−47	−15
21–30 yrs	3.5	43.3	1.8	34.1	1.3	31.9	−50	−21	−26	−6
>30 yrs	1.7	35.4	0.4	14.7	0.6	19.6	−75	−58	49	33
Education										
No education	9.9	56.4	7.6	50.1	3.9	43.6	−24	−11	−48	−13
Primary	4.8	42.0	3.8	40.9	2.0	33.1	−21	−3	−47	−19
Secondary	1.7	27.6	1.6	31.2	0.8	24.4	−6	13	−49	−22
Higher	0.3	12.7	0.2	9.6	0.2	14.7	−29	−25	15	54
Caste										
SC	10.3	57.4	6.1	50.0	2.5	38.8	−41	−13	−59	−22
ST	7.3	51.1	7.2	51.7	2.6	40.0	−2	1	−64	−23
Others	6.8	51.3	4.2	43.5	1.9	35.8	−39	−15	−55	−18
Religion										
Hindu	7.6	52.7	5.0	46.7	2.1	36.8	−34	−11	−58	−21
Muslim	7.5	51.8	4.5	42.3	2.3	39.0	−39	−18	−48	−8
Others	3.3	43.3	2.5	38.5	1.0	28.7	−24	−11	−59	−25
Wealth										
Poorest	10.4	57.2	8.7	53.0	4.3	44.4	−16	−7	−51	−16
Poorer	10.4	57.1	6.6	48.8	2.7	38.4	−36	−15	−59	−21
Middle	8.1	52.4	4.6	43.5	1.8	33.1	−43	−17	−61	−24
Richer	5.6	46.1	3.0	38.7	1.2	29.7	−46	−16	−62	−23
Richest	2.6	38.1	1.4	29.9	0.6	24.1	−45	−22	−58	−20
Residence										
Urban	4.2	45.2	2.8	39.9	1.2	31.1	−33	−12	−58	−22
Rural	8.4	53.7	5.7	47.4	2.5	38.4	−32	−12	−56	−19

Note: Table 3 is based on full birth history.

**Table 4 ijerph-19-14384-t004:** Simulation analysis of clustering of infant deaths in India and selected states, 2005–2006 and 2015–2016.

States/India	NFHS-3 (2005–2006)				Relative Change from the Existing Mortality Level NFHS-3 (in %)			NFHS-4 (2015–2016)				Relative Change from the Existing Mortality Level NFHS-4 (in %)		
	Existing Level (1a)	ID + 2CL (2a)	ID + 1CL (3a)	ID + 0CL (4a)	ID + 2CL (5a)	ID + 1CL (6a)	ID + 0CL (7a)	Existing Level (1b)	ID + 2CL (2b)	ID + 1CL (3b)	ID + 0CL (4b)	ID + 2CL (5b)	ID + 1CL (6b)	ID + 0CL (7b)
India	65	62	51	41	−5	−21	−37	41	41	35	30	−1	−14	−27
Uttarakhand	55	50	40	31	−9	−26	−44	42	41	35	29	−4	−18	−31
Rajasthan	73	67	55	40	−7	−25	−45	43	41	36	31	−3	−15	−28
Uttar Pradesh	83	77	62	46	−8	−25	−44	64	61	51	42	−6	−20	−35
Bihar	65	63	51	41	−4	−21	−37	48	46	39	32	−4	−18	−33
Assam	71	68	57	46	−4	−20	−34	48	46	40	33	−3	−17	−31
West Bengal	52	51	46	41	−2	−12	−21	31	31	27	23	−1	−13	−24
Jharkhand	77	73	58	45	−4	−24	−42	47	46	41	35	−2	−13	−26
Odisha	68	66	58	48	−2	−15	−29	45	43	37	33	−3	−16	−27
Chhattisgarh	81	76	63	49	−6	−22	−40	58	56	49	42	−4	−16	−29
Madhya Pradesh	82	76	60	45	−7	−27	−46	53	51	44	36	−5	−18	−32
Gujarat	63	62	52	43	−1	−17	−32	36	35	30	26	−2	−15	−26
Maharashtra	45	43	38	32	−5	−16	−29	24	24	22	20	−1	−8	−15
Kerala	18	18	15	14	0	−13	−21	7	6	6	5	−2	−7	−17
Tamil Nadu	38	38	33	30	0	−11	−21	20	20	18	17	−1	−8	−15

Note: # In the above Table 4, mortality estimation presented is based on 10-year BH; ID + 2CL: represents first situation where all women with multiple child loss in existing level of mortality situation (cols 1a and 1b) were randomly replaced by women who experienced exactly two child losses, and the estimated mortality is shown in col 2a and 2b; ID + 1CL: represents second situation where all women with multiple child loss in existing level of mortality situation (cols 1a and 1b) were randomly replaced by women who experienced exactly one child loss, and the estimated mortality is shown in cols 3a and 3b; ID + 1CL: represents third situation where all women with multiple child loss in existing level of mortality situation (cols 1a and 1b) were randomly replaced by women who experienced no child loss, and the estimated mortality is shown in cols 4a and 4b.

**Table 5 ijerph-19-14384-t005:** Intra-class correlation (ICC) and median odds ratio (MOR) and their relative change for India and selected states, 1992–2016.

States	NFHS-1 (1992–1993)				NFHS-3 (2005–2006)				NFHS-4 (2015–2016)				Relative Change NFHS-1 and NFHS-3 (in %)				Relative Change NFHS-3 NFHS-4 (in %)			
	Null Model		Model 1		Null Model		Model1		Null Model		Model 1		Null Model		Model 1		Null Model		Model 1	
	ICC	MOR	ICC	MOR	ICC	MOR	ICC	MOR	ICC	MOR	ICC	MOR	ICC	MOR	ICC	MOR	ICC	MOR	ICC	MOR
India	0.24	2.65	0.15	2.05	0.24	2.67	0.16	2.11	0.29	3.05	0.22	2.50	1.2	0.6	7.0	3.0	20.1	14.1	39.1	18.2
Uttarakhand	0.24	2.67	0.16	2.15	0.28	2.98	0.19	2.31	0.25	2.72	0.19	2.33	16.9	11.5	16.1	7.6	−11.8	−8.6	1.5	0.8
Rajasthan	0.32	3.26	0.23	2.57	0.21	2.41	0.10	1.80	0.28	2.90	0.21	2.42	−35.5	−26.0	−54.6	−29.9	34.2	20.5	99.0	34.4
Uttar Pradesh	0.16	2.15	0.09	1.74	0.14	2.04	0.07	1.63	0.22	2.49	0.15	2.06	−12.2	−5.3	−20.6	−6.5	50.6	22.1	100.2	26.3
Bihar	0.18	2.26	0.13	1.95	0.22	2.49	0.13	1.92	0.28	2.91	0.21	2.42	19.2	10.0	−4.1	−1.6	27.3	17.1	65.5	25.8
Assam	0.18	2.26	0.11	1.84	0.31	3.18	0.24	2.62	0.31	3.21	0.25	2.68	70.7	40.9	114.9	42.7	1.0	0.9	3.4	2.2
West Bengal	0.16	2.13	0.08	1.69	0.24	2.61	0.16	2.12	0.25	2.72	0.16	2.15	46.6	22.6	86.1	24.9	6.3	4.2	4.1	1.8
Jharkhand	0.29	3.03	0.20	2.41	0.18	2.25	0.09	1.70	0.26	2.79	0.18	2.22	−37.6	−25.5	−58.0	−29.4	43.8	23.8	103.5	30.6
Odisha	0.15	2.08	0.08	1.66	0.18	2.28	0.10	1.78	0.25	2.74	0.17	2.17	21.1	9.5	25.9	7.1	37.9	20.5	67.6	22.0
Chhattisgarh	0.30	3.10	0.22	2.49	0.21	2.42	0.11	1.85	0.26	2.82	0.19	2.32	−31.1	−22.0	−48.4	−25.6	28.5	16.9	70.7	25.3
Madhya Pradesh	0.25	2.68	0.18	2.23	0.20	2.39	0.11	1.86	0.26	2.82	0.18	2.26	−17.1	−10.6	−35.3	−16.4	29.9	17.7	59.3	21.4
Gujarat	0.21	2.43	0.14	2.00	0.22	2.49	0.13	1.95	0.34	3.48	0.26	2.78	4.3	2.5	−5.5	−2.3	56.6	39.3	99.4	42.8
Maharashtra	0.28	2.97	0.18	2.24	0.32	3.23	0.20	2.35	0.37	3.78	0.28	2.95	11.3	8.8	9.8	4.8	17.9	17.0	43.2	25.5
Kerala	0.24	2.67	0.15	2.06	0.39	3.97	0.36	3.70	0.51	5.75	0.46	4.88	59.0	48.3	145.9	79.8	30.3	45.0	25.4	32.1
Tamil Nadu	0.21	2.42	0.12	1.92	0.25	2.68	0.19	2.29	0.41	4.21	0.33	3.38	18.4	10.8	49.0	19.0	66.6	57.0	78.3	47.9

**Table 6 ijerph-19-14384-t006:** Results of random effects dynamic probit model decomposition analysis of infant mortality into factors, India, 1992 and 2006.

Explanatory Variables	NFHS-1		NFHS-3		Proportion of Overall Change Due to Change in Rates, Comp., and Interaction		
	P1	B1	P2	B2			
					Rates	Composition	Interaction
Previous Death							
No ^®^							
Yes	4.95	0.25 ***	3.41	0.29 ***	14.5	−22.9	−4.5
Total					14.5	−22.9	−4.5
Mother’s Age							
15–29 ^®^							
30–39	38.19	0.00	38.41	−0.02	−45.8	0.0	−0.3
40–49	7.60	0.16 ***	6.10	0.19 ***	13.2	−14.6	−2.6
Total					−32.6	−14.6	−2.9
Region							
Central ^®^							
North	22.94	−0.18 ***	15.62	−0.11 ***	104.2	82.1	−33.2
East	17.04	−0.07 ***	15.52	−0.12 ***	−50.4	6.6	4.5
Northeast	11.91	−0.24 ***	18.61	−0.16 ***	56.2	−95.7	31.7
West	10.59	−0.31 ***	10.55	−0.15 ***	103.9	0.7	−0.4
South	15.04	−0.25 ***	14.40	−0.25 ***	2.9	9.9	−0.1
Total					216.8	3.5	2.4
Residence							
Rural ^®^							
Urban	27.39	0.02	37.76	0.07 **	74.9	15.7	28.3
Total					74.9	15.7	28.3
Mother’s Education							
Higher ^®^							
Illiterate	63.05	0.56 ***	45.98	0.51 ***	−202.5	−580.2	54.8
Primary	16.06	0.49 ***	14.48	0.47 ***	−16.4	−47.0	1.6
Secondary	17.87	0.36 ***	33.20	0.32 ***	−37.6	332.0	−32.3
Total					−256.5	−295.3	24.1
Religion							
Hindu ^®^							
Others	24.08	−0.14	30.86	−0.05	123.9	−57.3	34.9
Total					123.9	−57.3	34.9
Caste							
Other caste ^®^							
SC/ST	26.26	−0.02	33.94	0.00	42.5	−11.6	12.4
Total					42.5	−11.6	12.4
Wealth Index							
Poorest ^®^							
Poorer	18.09	−0.07 ***	18.94	−0.01	61.3	−3.6	2.9
Middle	21.37	−0.12 ***	20.69	−0.09 **	39.6	4.9	−1.3
Richer	23.23	−0.23 ***	21.54	−0.13 ***	130.5	23.2	−9.5
Richest	19.68	−0.31 ***	20.23	−0.25 ***	70.1	−10.4	2.0
Total					301.5	14.1	−5.9
Child’s Sex							
Male ^®^							
Female	48.25	0.03 *	47.97	0.00	−97.8	−0.5	0.6
Total					−97.8	−0.5	0.6
Intercept		−1.83 ***		−1.96 ***	−7.72		
Grand Total					379.4	−368.9	89.5
Number of Observations	113,971		89,625				
Rho(ICC)	0.159		0.135				
Mother-level unobserved heterogeneity	0.6212 ***	0.022	0.514 ***	0.032			
Theta	0.711 ***	0.105	0.857 ***	0.197			

Note: ^®^ denotes reference category, *** *p* < 0.01; ** *p* < 0.05; * *p* < 0.1; based on 10-year birth history.

**Table 7 ijerph-19-14384-t007:** Result of random effects dynamic probit model decomposition analysis of infant mortality into factors, India, 2006 and 2016.

Explanatory Variables	NFHS-3		NFHS-4		Proportion of Overall Change Due to Change in Rates, Comp., and Interaction		
	P1	B1	P2	B2			
					Rates	Composition	Interaction
Previous Death							
No ^®^							
Yes	3.41	0.293 ***	2.60	0.239 ***	1.15	1.48	−0.27
Total					1.15	1.48	−0.27
Mother’s Age							
15–29 ^®^							
30–39	38.41	−0.020	42.18	0.033 ***	−12.79	0.48	−1.26
40–49	6.10	0.188 ***	7.70	0.220 ***	−1.21	−1.87	−0.32
Total					−14.00	−1.39	−1.57
Region							
Central ^®^							
North	15.62	−0.110 ***	18.77	−0.181 ***	6.90	2.15	1.39
East	15.52	−0.120 ***	21.11	−0.197 ***	7.39	4.17	2.66
Northeast	18.61	−0.157 ***	15.44	−0.210 ***	6.09	−3.10	−1.04
West	10.55	−0.146 ***	6.79	−0.342 ***	12.87	−3.40	−4.58
South	14.40	−0.250 ***	8.78	−0.354 ***	9.30	−8.75	−3.63
Total					42.55	−8.93	−5.20
Residence							
Rural ^®^							
Urban	37.76	0.070 **	23.03	0.017	12.35	6.41	−4.82
Total					12.35	6.41	−4.82
Mother’s Education							
Higher ^®^							
Illiterate	45.98	0.507 ***	37.96	0.318 ***	53.87	25.28	−9.40
Primary	14.48	0.474 ***	15.44	0.280 ***	17.46	−2.82	1.15
Secondary	33.20	0.322 ***	39.65	0.211 ***	23.00	−12.91	4.47
Total					94.32	9.55	−3.78
Religion							
Hindu ^®^							
Others	30.86	−0.054	28.60	−0.055 ***	0.18	−0.76	−0.01
Total					0.18	−0.76	−0.01
Caste							
Other caste ^®^							
SC_ST	33.94	0.002	39.42	0.010	−1.71	−0.06	−0.28
Total					−1.71	−0.06	−0.28
Wealth Index							
Poorest ^®^							
Poorer	18.94	−0.014	24.04	−0.017	0.28	0.45	0.08
Middle	20.69	−0.089 **	19.62	−0.043 **	−5.89	−0.59	0.31
Richer	21.54	−0.133 ***	15.82	−0.068 ***	−8.82	−4.74	2.34
Richest	20.23	−0.251 ***	12.44	−0.144 ***	−13.47	−12.15	5.19
Total					−27.90	−17.03	7.91
Child’s Sex							
Male ^®^							
Female	47.97	−0.002	47.94	−0.043 ***	12.10	0.00	−0.01
Total					12.10	0.00	−0.01
Intercept		−1.961 ***		−1.920 ***	−0.26		
Grand Total					118.78	−10.74	−8.03
Number of Observations	89,625		462,507				
Rho (Intra-class correlation coefficient)	0.135		0.182				
Mother-level unobserved heterogeneity	0.514 ***	0.032	0.732 ***	0.016			
Theta	0.857 ***	0.197	0.844	0.075			

Note: ^®^ denotes reference category, *** *p* < 0.01; ** *p* < 0.05; based on 10-year birth history.

**Table 8 ijerph-19-14384-t008:** Socio-demographic characteristics of women with no child loss, women with exactly one child loss, and women with loss of two or more children, India, 1992–2016.

Covariates	Child Loss in NFHS-1 (1992–1993)			Child Loss in NFHS-3 (2005–2006)			Child Loss in NFHS-4 (2015–2016)		
	No	One	Two or More	No	One	Two or More	No	One	Two or More
Residence									
Urban	23.47	15.88	16.15	26.08	18.95	13.96	28.66	20.17	12.88
Rural	76.53	84.12	83.85	73.92	81.05	86.04	71.34	79.83	87.12
Education									
No education	64.01	74.58	82.76	48.79	61.52	67.23	29.36	38.72	49.48
Primary	14.76	13.29	11.02	13.97	14.83	15.72	13.85	17.63	16.65
Secondary	18.28	11.07	6.21	31.96	21.71	17.07	45.96	38.82	30.15
Higher	2.95	1.06	0.00	5.27	1.95	0.00	10.83	4.86	3.72
Media									
TV (weekly)									
No	72.99	81.01	81.92	45.09	54.74	61.34	30.31	40.37	50.62
Yes	27.01	18.99	18.08	54.91	45.26	38.66	69.72	59.63	49.43
Radio (weekly)									
No	60.81	67.19	65.55	60.42	61.66	62.31	86.52	87.77	85.25
Yes	39.19	32.81	34.45	39.63	38.34	37.69	13.48	12.23	14.75
Sanitation Facilities									
Not Improved	73.92	84.67	91.81	30.58	21.06	16.69	46.96	37.14	26.15
Improved	26.08	15.33	8.19	60.83	69.69	73.57	46.86	56.53	67.48
De jure	NA	NA	NA	8.59	9.26	9.75	6.18	6.33	6.36
Religion									
Hindu	79.14	82.84	80.07	78.06	79.67	84.22	78.55	80.37	79.72
Muslims	15.47	14.38	16.42	17.18	16.75	10.08	16.58	16.32	18.6
Others	5.39	2.78	3.51	4.76	3.57	5.70	4.87	3.31	1.68
Caste									
SC	12.91	17.24	17.93	20.49	23.76	30.62	21.4	23.83	27.24
ST	9.54	9.91	9.79	9.42	10.96	9.46	10.49	11.06	9.85
Others	77.55	72.85	72.28	70.09	65.28	59.93	68.11	65.11	62.91
Parity									
1st	20.43	12.55	0.00	19.95	8.67	0.00	26.33	10.12	0.00
2nd	26.01	27.82	18.81	30.83	29.17	19.87	38.38	32.47	19.13
3rd	19.68	19.59	29.23	19.19	23.21	13.79	18.58	28.77	32.82
4 or more	33.88	40.06	51.96	30.03	38.95	66.35	16.71	28.64	48.04
Mother’s age (at birth)									
12–20 yrs	30.72	36.22	37.85	28.79	35.29	31.39	21.58	24.55	21.69
21–30 yrs	56.43	51.02	42.84	59.77	54.19	60.27	67.73	64.04	62.29
31–49 yrs	12.87	12.76	19.32	11.44	10.51	8.35	10.69	11.42	16.03
Mother’s Work Status									
Not working	72.76	70.77	72.79	63.18	58.46	52.35			
Working	27.24	29.23	27.21	36.82	41.54	47.65			
Total	43,522	4363	370	46,496	3821	385	237,738	15,829	1371

**Table 9 ijerph-19-14384-t009:** Utilization of health services of women with no child loss, women with exactly one child loss, and women with loss of two or more children, India, 1992–2016.

Covariates	Child Loss in NFHS-1 (1992–1993)			Child Loss in NFHS-3 (2005–2006)			Child Loss in NFHS-4 (2015–2016)		
	No	One	Two or More	No	One	Two or More	No	One	Two or More
ANC									
No	36.02	49.58	60.72	23.16	28.06	40.86	17.03	22.96	31.87
1–3	36.06	32.44	27.14	39.27	43.73	33.98	31.22	37.35	38.29
4 or more	27.92	17.97	12.16	37.56	28.21	25.15	51.81	39.72	29.84
IFA									
No	48.06	61.10	70.11	8.56	8.21	12.28	2.14	1.96	2.32
Yes	51.94	38.90	29.89						
less than 100	NA	NA	NA	67.94	72.51	63.37	58.56	66.17	64.18
100 or more	NA	NA	NA	23.51	19.28	24.35	39.30	31.86	33.51
TT Injection									
No	37.51	51.88	62.48	16.59	22.71	28.76	8.59	11.11	12.64
One	7.11	7.60	5.98	6.65	7.34	6.27	8.14	10.64	13.76
2 or more	55.38	40.51	31.54	76.76	69.94	64.97	83.27	78.29	73.63
Place of Delivery									
Home									
Institutional	26.29	17.58	11.99	39.43	30.68	26.58	79.63	72.18	66.61
Full Immunization									
No									
Yes	35.59	11.12	1.88	43.55	19.01	7.41	61.86	30.46	9.83
Total	43,522	4363	370	46,496	3821	385	237,738	15,829	1371

Note: antenatal care (ANC), iron folic acid (IFA), tetanus toxoid injection (TT), full immunization for child aged 12–23 months.

## Data Availability

The data are available in the public domain and can be accessed by https://dhsprogram.com/data/available-datasets.cfm (20 August 2020).

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
