# Peer review of "Infant Death Clustering in the Quarter of a Century in India: A Decomposition Analysis"

_ijerph, 2022, doi:10.3390/ijerph192114384_

Round 1

Reviewer 1 Report

The study is examining the clustering of deaths of infant children in India and how socio-demographic and economic factors explain the decline in infant deaths. In my opinion this study is a great value addition in the aforesaid topic as India is still facing a huge burden of infant deaths and a significant component of total infant deaths is clustering of the death. So analyzing the factors contributing in this phenomenon will be of great importance for the policymakers and researchers.

I just would like to draw the authors kind attention towards one number in table 8. In NFHS4, the mothers who lost two child, among them the percent of Institutional delivery is 67%. So it will be highly appreciable if you can further strengthen your discussion section by throwing some light on the quality of institutional deliveries and the care provided by the institution.

I have few small observations which should be corrected

1) In the categories of education, there is category of illiterate, I would suggest to use the phrase "No formal education"

2) In the table 8, there are variables which are having only two categories and are compliment to each other (i.e together they constitute 100%), like IFA, place of delivery, immunisation. I would suggest to write only one figure, other will be self explanatory.

Author Response

Reviewer 1
Comment 1: In the categories of education, there is category of illiterate, I would suggest to use the phrase "No formal education"
Response: In India “illiterate” and “No formal education” have different connotations. Illiterate definition as per census of India is those who cannot read, write, and speak at least one language. However, “No formal education” does not necessarily mean that one cannot read, write and speak.
Comment 2: In the table 8, there are variables which are having only two categories and are compliment to each other (i.e., together they constitute 100%), like IFA, place of delivery, immunization. I would suggest to write only one figure, other will be self-explanatory.
Response: We agree with the reviewer and changed it accordingly.

Reviewer 2 Report

Dear Authors,

I read your article with interest. However, I have suggestions for necessary corrections:

1. Unfortunately, the abstract does not fully describe what the article is about, what methods were used and what results were achieved;

2. In Introduction (and then in Discussion) it is necessary to mention the pioneers of research on mortality, including children, i.e. A. Omran, McKeown and especially Caldwell. In the attachment (see Box. 9.1) I am sending the article:

Lecka I., 2018. Geographies of twenty-first-century disease: epidemiological versus demographic transition. In: M. W. Solarz (ed.) New Geographies of the Globalized World. London, Rutledge: 187-211. 

3. You need to improve the style, a little more clearly (dividing into paragraphs) to write the Introduction. See e.g. lines 48-53 and the like.

4. Standardize the data given in lines 69 and 90 and 93 (different time intervals ...).

5. Subsections 2.1; 2,2; 2,3 and chapter 3 are not preceded by a few introductory sentences. We don't start with a formula or a table.

6. The result is described in a block-like manner. More story grouping paragraphs are required.

7. The chapter Discussion  actually need introducing a discussion with other studies that need to be quoted.

8. National Health Mission (line 321) no quotation, we don't know what was put into practice in this project. Why do we only find out about programm here? and why only one, even if it's a flagship .... Too many mental shortcuts.

9. The topic linking the increase in women's education with the increase in mortality is worth considering for a while (see Caldwell et al.).

Summing up, the presented article is a dry description of the statistical method used, but there is no value for Implications, because it treats all the counted categories briefly. So in fact, apart from the statement of increase-decrease in mortality plus the well-known clustering, we know nothing about the reality of the causes /factors/circumstances. It needs to be somehow fixed.

I will be happy to read the article revised soon. The amendments mainly concern a clearer description, refinement of the Discussion, introduction of more citations of studies with which the polemics are conducted.

Author Response

Reviewer 2
Comment 1. Unfortunately, the abstract does not fully describe what the article is about, what methods were used and what results were achieved;
Response: The abstract is revised to address the reviewer’s comments. Comment 2. In Introduction (and then in Discussion) it is necessary to mention the pioneers of research on mortality, including children, i.e. A. Omran, McKeown and especially Caldwell. In the attachment (see Box. 9.1) I am sending the article: Lecka I., 2018. Geographies of twenty-first-century disease: epidemiological versus demographic transition. In: M. W. Solarz (ed.) New Geographies of the Globalized World. London, Rutledge: 187-211.
Response: We included these important references in the introduction as well as in the discussion as per the reviewer.
Comment 3. You need to improve the style, a little more clearly (dividing into paragraphs) to write the Introduction. See e.g. lines 48-53 and the like.
Response: Introduction is revised to reflect the reviewer’s comment. 

Comment 4. Standardize the data given in lines 69 and 90 and 93 (different time intervals ...).
Response :We have standardized it.

Comment 5. Subsections 2.1; 2,2; 2,3 and chapter 3 are not preceded by a few introductory sentences. We don't start with a formula or a table.
Response :We added a few introductory sentences before introducing a formula.

Comment 6. The result is described in a block-like manner. More story grouping paragraphs are required.
Response: The results’ section is revised to reflect the reviewer’s comments.

Comment 7. The chapter Discussion actually need introducing a discussion with other studies that need to be quoted.
Response: We have tried to improve the discussion.

Comment 8. National Health Mission (line 321) no quotation, we don't know what was put into practice in this project. Why do we only find out about programme here? and why only one, even if it's a flagship .... Too many mental shortcuts.
Response: We thank the reviewer for the comment and tried to address it.

Comment 9. The topic linking the increase in women's education with the increase in mortality is worth considering for a while (see Caldwell et al.).
Response: We have addressed the reviewer’s comment in the discussion section.

Reviewer 3 Report

This is a very thoughtful topic; it is helpful not only for demographic assessment but also as a tool for identifying avenues for improvement in the future. 

Some specific questions I had while reading the manuscript are: 

#42: the meaning is not clear

#97: Please elucidate 'Kids File'

#321: ‘been’ needs to be removed 

#329: Please check the spelling of the program; I believe it should be spelled as “Janani Suraksha Yojana,” and please explain specifics of how it helped. 

#354: Please explain “UP TSU”

#409: ‘correlate with’ instead of ‘corroborate’?

Some sentences may need to be reworded as can be identified copied directly from a previously published manuscript (Ranjan M, Dwivedi LK. Death clustering in India: levels, trends, and differentials, 1992–2016. Indian Journal of Child Health. 2019 Apr 29;6(4):165-72), https://mansapublishers.com/IJCH/article/view/1451. Since that manuscript was published recently, with similar data, it may be prudent to ensure this manuscript stands out in comparison. In addition, I highly recommend the manuscript undergo a detailed revision to review grammatical and spelling errors. 

Author Response

Reviewer 3
This is a very thoughtful topic; it is helpful not only for demographic assessment but also as a tool for identifying avenues for improvement in the future.
Some specific questions I had while reading the manuscript are:
Comment 1. #42: the meaning is not clear
Response: We think now it is clear.
Comment 2. #97: Please elucidate 'Kids File'
Response: We added the details about the Kids file in the data section
Comment 3. #321: ‘been’ needs to be removed
Response: It has been removed.
Comment 4. #329: Please check the spelling of the program; I believe it should be spelled as “Janani Suraksha Yojana,” and please explain specifics of how it helped.
Response: We thank the reviewer and revised accordingly.

Comment 5. #354: Please explain “UP TSU”
Response: UP TSU stands for Uttar Pradesh Technical Support Unit.

Round 2

Reviewer 2 Report

Dear Authors,

thank you for your work. I see your article as a much more clear and suitable to be published.

Reviewer 3 Report

Thank you for the updates and edits that have significantly enhanced the manuscript's content. However, it still needs an extensive revision of grammar and language.